# Deep Active Inference Agents for Delayed and Long-Horizon Environments

## Abstract

With the recent success of *world-model* agents—which extend the core idea of model-based reinforcement learning by learning a differentiable model for sample-efficient control across diverse tasks—*active inference* (AIF) offers a complementary, neuroscience-grounded paradigm that unifies perception, learning, and action within a single probabilistic framework powered by a generative model. Despite this promise, practical AIF agents still rely on accurate *immediate* predictions and exhaustive planning, a limitation that is exacerbated in *delayed* environments requiring planning over *long horizons*—tens to hundreds of steps. Moreover, most existing agents are evaluated on robotic or vision benchmarks which, while natural for biological agents, fall short of real-world industrial complexity. We address these limitations with a generative–policy architecture featuring (i) a *multi-step latent transition* that lets the generative model predict an entire horizon in a single look-ahead, (ii) an integrated policy network that enables the transition and receives gradients of the expected free energy, (iii) an alternating optimization scheme that updates the model and policy from a replay buffer, and (iv) a single gradient step that plans over long horizons, eliminating exhaustive planning from the control loop. We evaluate our agent in an environment that mimics a realistic industrial scenario with delayed and long-horizon settings. The empirical results confirm the effectiveness of the proposed approach, demonstrating the coupled world-model with the AIF formalism yields an end-to-end probabilistic controller capable of effective decision making in delayed, long-horizon settings without handcrafted rewards or expensive planning.

## 1 Introduction

There has been significant progress in data-driven decision-making algorithms, particularly in reinforcement learning (RL), where agents learn policies through interaction with the environment and receive feedback (Sutton & Barto, 2018). Deep learning, in parallel, offers a powerful framework for extracting representations and patterns, while also enabling probabilistic modeling (LeCun et al., 2015; Bishop & Bishop, 2024), driving advancements in computer vision, natural language processing, biomedical applications, finance, and robotics. Deep RL merges these ideas—for example, by using neural function approximation in Deep Q-Networks (DQN), which achieved human-level performance on Atari games (Mnih et al., 2015). Model-based RL (MBRL) goes further by explicitly incorporating a model—either learned or provided—of the environment to guide learning and planning (Moerland et al., 2023). Similarly, the concept of world models centers on learning generative models of the environment to exploit representations and predictions of future outcomes, especially for decision-making (Hafner et al., 2025), and, in fact, general agents are provably required to contain internal world models (Richens et al., 2025). This resonates with cognitive theories of the biological brain, which emphasize the role of internal generative models (Friston et al., 2021). At a broader theoretical level, active inference (AIF), an emerging field in neuroscience, unifies perception, action, and learning in biological agents through the use of internal generative models (Friston et al., 2017; Parr et al., 2022).

AIF is grounded in the free energy principle (FEP), which formulates neural inference and learning under uncertainty as minimization of *surprise* (Friston, 2010). It provides a coherent mathematical framework that calibrates a probabilistic model governed by Bayesian inference, enabling both learning and goal-directed action directly from raw sensory inputs (i.e., *observations*) (Parr et al.,

2022). This can support the development of model-driven, adaptive agents that are trained end-to-end while offering uncertainty quantification and some interpretability (Taheri Yeganeh et al., 2024; Fountas et al., 2020). Similar to world models and model-based RL, AIF is powered by an internal model of the environment, which can help to capture dynamics and boost sample efficiency. Despite the potential of the AIF framework, its practical agents typically rely on accurate immediate predictions and extensive planning (Fountas et al., 2020). Such reliance can hinder performance, particularly in *delayed* environments, where the consequences of actions are not immediately observable—commonly framed in RL as *sparse rewards*, which exacerbates the credit-assignment problem (Sutton & Barto, 2018). Likewise, *long-horizon* tasks demand effective planning over extended temporal horizons, posing an additional challenge. These difficulties appear across diverse optimization tasks—such as manufacturing systems (Taheri Yeganeh et al., 2024), robotics (Hafner et al., 2020; 2025; Nguyen et al., 2024), and protein design (Angermueller et al., 2019; Wang et al., 2024)—where the consequences become apparent only after many steps or upon completion of the entire process.

We explore how the potential of the AIF framework can be harnessed to build agents that remain effective in environments that are delayed and demanding long-horizon planning. Recent advances in deep generative modeling (Tomczak, 2024) have unlocked breakthroughs across diverse domains—such as AlphaFold's high-accuracy protein-structure predictions (Abramson et al., 2024). Because the generative model is the core of AIF, our objective is to extend its capacity and fidelity as the world model by predicting deep into the future. Concretely, we propose a generative model with an integrated policy network, trained end-to-end under the AIF formalism, allowing the model to produce long-horizon roll-outs and supply gradient signals to the policy during optimization. The summary of our contributions is as follows:

- We introduce an AIF-consistent generative–policy architecture that enables long-horizon predictions while providing differentiable signals to the policy.

- We derive a joint training algorithm that alternately updates the generative model and the policy network, and we show how the learned model can be leveraged during planning via gradient updates to the policy.

- We empirically demonstrate the concept's effectiveness in an industrial environment, highlighting its relevance to delayed and long-horizon scenarios.

The remainder of the paper is organized as follows: Section 2 reviews the formalism and planning strategies. Section 3 presents our proposed concept and agent architecture, while Section 4 details the experimental results. Finally, Section 5 concludes with implications and outlines future directions.

## 2 BACKGROUND

Agents based on the world models concept extend the core idea of MBRL, learning a differentiable predictive model to facilitate policy optimization and planning via *imaginations* in the model (Ha & Schmidhuber, 2018; Hafner et al., 2025). They create latent representations that capture spatial and temporal aspects to model dynamics and predict the future (Ha & Schmidhuber, 2018). The architecture governing this dynamics—generative model—and how it is leveraged for policy and planning is foundational in this concept. Many designs resemble variational autoencoder (Kingma & Welling, 2013) and are often augmented with Recurrent State-Space Models (RSSMs) to provide memory and help with credit assignment (Hafner et al., 2019; 2025; Nguyen et al., 2024). At the same time, RL methods such as actor–critic (Sutton & Barto, 2018) are integrated with the model to optimize the policy (Hafner et al., 2020; 2025; Nguyen et al., 2024), yielding sample-efficient agents that rely on imagination rather than extensive environment interaction.

AIF offers a complementary, neuroscience-grounded perspective that subsumes predictive coding that postulates that the brain minimizes prediction errors—relative to an internal generative model of the world—under uncertainty (Millidge et al., 2022). It casts the brain as a hierarchy that performs variational Bayesian inference continuously to suppress prediction error (Parr et al., 2022). It was originally advanced to explain how organisms actively control and navigate their environments by iteratively updating beliefs and inferring actions from sensory observations (Parr et al., 2022). AIF emphasizes the dependency of observations on actions (Millidge et al., 2022); accordingly, it posits

that actions are chosen, while calibrating the model, to align with preferences and reduce uncertainty, thereby unifying perception, action, and learning (Millidge et al., 2022). The free-energy principle provides the mathematical bedrock for this framework (Friston et al., 2010; Millidge, 2021), and a growing body of empirical work supports its biological plausibility (Isomura et al., 2023). AIF-based agents have been deployed in robotics, autonomous driving, and clinical decision support (Pezzato et al., 2023; Schneider et al., 2022; Huang et al., 2024), demonstrating robust performance in uncertain, dynamic settings. In this work, we adopt the AIF formulation of Fountas et al. (2020), which was extended in (Da Costa et al., 2022; Taheri Yeganeh et al., 2024) and has been shown to result in effective agents across different environments—such as visual and industrial tasks.

## 2.1 FORMALISM

Within AIF, agents employ an integrated probabilistic framework consisting of an internal generative model (Da Costa et al., 2023) with inference mechanisms that allow them to represent and act upon the world. The framework assumes a Partially Observable Markov Decision Process (Kaelbling et al., 1998; Da Costa et al., 2023; Paul et al., 2023), where an agent's interaction with its environment is formalized in terms of three random variables—observation, latent state, and action—denoted $(o_t, s_t, a_t)$ at time $t$. In contrast to RL, this formalism does not rely on explicit reward feedback from the environment; instead, the agent learns solely from the sequence of observations it receives. The agent's generative model, parameterized by $\theta$, is defined over trajectories as $P_\theta(o_{1:t}, s_{1:t}, a_{1:t-1})$ up to time $t$. The agent's behavior is driven by the imperative to minimize *surprise*, which is formulated as the negative log-evidence for the current observation, $-\log P_\theta(o_t)$ (Fountas et al., 2020). The agent approaches this imperative from two perspectives when interacting with the world, as follows (Parr et al., 2022; Fountas et al., 2020):

1) Using the current observation, the agent calibrates its generative model by optimizing the parameters $\theta$ to yield more accurate predictions. Mathematically, this surprise can be expanded as follows (Kingma & Welling, 2013):

$$-\log P_\theta(o_t) \leq \mathbb{E}_{Q_\phi(s_t, a_t)}\left[\log Q_\phi(s_t, a_t) - \log P_\theta(o_t, s_t, a_t)\right] , \tag{1}$$

which provides an upper bound, commonly known as the negative Evidence Lower Bound (ELBO) (Blei et al., 2017). It is widely used as a loss function for training variational autoencoders (Kingma & Welling, 2013). In AIF, it corresponds to the Variational Free Energy (VFE), whose minimization reduces the surprise associated with predictions relative to actual observations (Fountas et al., 2020; Sajid et al., 2022; Paul et al., 2023).

2) Looking into the future, where the agent needs to plan actions, an estimate of the surprise of future predictions can be obtained. Considering a sequence of actions—or policy—denoted as $\pi$, for $\tau \geq t$, this corresponds to $-\log P(o_\tau|\theta, \pi)$, which can be estimated analogously to VFE (Schwartenbeck et al., 2019):

$$G(\pi, \tau) = \mathbb{E}_{P(o_\tau|s_\tau, \theta)}\mathbb{E}_{Q_\phi(s_\tau, \theta|\pi)}\left[\log Q_\phi(s_\tau, \theta|\pi) - \log P(o_\tau, s_\tau, \theta|\pi)\right] . \tag{2}$$

This is known as the Expected Free Energy (EFE) in the framework, which quantifies the relative quality of policies—lower values correspond to better policies.

The EFE in Eq. 2 can be derived as a decomposition of distinct terms for time $\tau$, as follows (Schwartenbeck et al., 2019; Fountas et al., 2020):

$$G(\pi, \tau) = -\mathbb{E}_{\tilde{Q}}\left[\log P(o_\tau|\pi)\right] \tag{3a}$$

$$+ \mathbb{E}_{\tilde{Q}}\left[\log Q(s_\tau|\pi) - \log P(s_\tau|o_\tau, \pi)\right] \tag{3b}$$

$$+ \mathbb{E}_{\tilde{Q}}\left[\log Q(\theta|s_\tau, \pi) - \log P(\theta|s_\tau, o_\tau, \pi)\right] , \tag{3c}$$

where $\tilde{Q} = Q(o_\tau, s_\tau, \theta|\pi)$. Fountas et al. (2020) rearranged this formulation with further use of sampling, leading to a tractable estimate for the EFE that is both interpretable and easy to calculate (Fountas et al., 2020):

$$G(\pi, \tau) = -\mathbb{E}_{Q(\theta|\pi)Q(s_\tau|\theta, \pi)Q(o_\tau|s_\tau, \theta, \pi)}\left[\log P(o_\tau|\pi)\right] \tag{4a}$$

$$+ \mathbb{E}_{Q(\theta|\pi)}\left[\mathbb{E}_{Q(o_\tau|\theta, \pi)}H(s_\tau|o_\tau, \pi) - H(s_\tau|\pi)\right] \tag{4b}$$

$$+ \mathbb{E}_{Q(\theta|\pi)Q(s_\tau|\theta, \pi)}H(o_\tau|s_\tau, \theta, \pi) - \mathbb{E}_{Q(s_\tau|\pi)}H(o_\tau|s_\tau, \pi) . \tag{4c}$$

Conceptually, the contribution of each term in the EFE can be interpreted as follows (Fountas et al., 2020): Extrinsic value (Eq. 4a) — the expected *surprise*, which measures the mismatch between the outcomes predicted under policy $\pi$ and the agent's prior preferences over outcomes. This term is analogous to reward in RL, as it quantifies the misalignment between predicted and preferred outcomes. However, rather than maximizing cumulative reward, the agent minimizes surprise relative to preferred observations. State epistemic uncertainty (Eq. 4b) — mutual information between the agent's beliefs about states before and after obtaining new observations. This term incentivizes exploration of regions in the environment that reduce uncertainty about latent states (Fountas et al., 2020). Parameter epistemic uncertainty (Eq. 4c) — the expected information gain about model parameters given new observations. This term also corresponds to active learning or curiosity (Fountas et al., 2020), and reflects the role of model parameters $\theta$ in generating predictions. The last two terms capture distinct forms of epistemic uncertainty, providing an intrinsic drive for the agent to explore and refine its generative model. They play a role analogous to intrinsic rewards in RL that balance the exploration–exploitation trade-off. Similar information-seeking or curiosity signals underpin many successful RL algorithms—ranging from curiosity-driven bonuses (Pathak et al., 2017; Burda et al., 2018) to the entropy-regularized objective optimized by Soft Actor-Critic (Haarnoja et al., 2018)—and have been shown to yield strong, sample-efficient agents.

## 2.2 PLANNING STRATEGY

Agents based on MBRL typically leverage their world model to *imagine* future trajectories before acting, trading extra computation for large gains in sample efficiency and performance. Monte Carlo Tree Search (MCTS) (Coulom, 2006; Silver et al., 2017) is a notable search algorithm, which selectively explores promising trajectories in a restricted manner. Its effectiveness was highlighted in *AlphaGo Zero* (Silver et al., 2017) and later by *MuZero*, which folds a learned latent dynamics model directly into the search loop (Schrittwieser et al., 2020). In the AIF concept, the agent's planning before taking actions is to minimize the EFE; mathematically, it corresponds to the negative accumulated EFE $G$ as follows:

$$P(\pi) = \sigma(-G(\pi)) = \sigma \left( - \sum_{\tau > t} G(\pi, \tau) \right) , \qquad (5)$$

where $\sigma(\cdot)$ represents the *Softmax* function. The agent simulates possible trajectories via roll-outs from its generative model, under policy $\pi$, to evaluate the EFE. However, calculating this for any possible $\pi$ is infeasible as the policy space grows exponentially with the depth of planning. Fountas et al. (2020) introduced an auxiliary module along with the MCTS to alleviate this obstacle. They proposed a recognition module (Piché et al., 2018; Marino et al., 2018; Tschantz et al., 2020), parameterized with $\phi_a$ as follows: *Habit*, $Q_{\phi_a}(a_t)$, which approximates the posterior distribution over actions using the prior $P(a_t)$ that is returned from the MCTS (Fountas et al., 2020). This is similar to the fast and habitual decision-making in biological agents (Van Der Meer et al., 2012). They used this module for fast expansions of the search tree during planning, followed by calculating the EFE of the leaf nodes and backpropagating over the trajectory. Iteratively, it results in a weighted tree with memory updates for the visited nodes. They also used the Kullback–Leibler divergence between the planner's policy and the habit as precision to modulate the latent state (Fountas et al., 2020). Another approach to enhance the planning is using a *hybrid horizon* (Taheri Yeganeh et al., 2024), in which the short-sighted EFE terms—based on immediate next-step predictions—are augmented with an additional term during planning to account for longer horizons. Taheri Yeganeh et al. (2024) employed a Q-value network, $Q_{\phi_a}(a_t)$, to represent the amortized inference of actions, trained in a model-free manner using extrinsic values. These terms were then combined in the planner as follows:

$$P(a_t) = \gamma \cdot Q_{\phi_a}(a_t) + (1 - \gamma) \cdot \sigma \left( -G(\pi) \right) , \qquad (6)$$

balancing long-horizon extrinsic value against short-horizon epistemic drive.

Modern world-model agents increasingly shift the look-ahead into latent space; PlaNet (Hafner et al., 2019) uses cross-entropy method roll-outs inside a RSSM trained with *latent overshooting*, while the Dreamer family (Hafner et al., 2020; 2025) propagates analytic value gradients through hundreds of imagined trajectories, without a tree search. EfficientZero (Ye et al., 2021) blends AlphaZero-style MCTS with latent-space imagination, surpassing human Atari performance with only 100k frames. These approaches typically couple multi-step model roll-outs with an actor (policy) and often a critic (value) network that are queried during imagination. In each simulated step,

the policy proposes the next action and the critic supplies a bootstrapped value, enabling efficient multi-step look-ahead without enumerating the full action tree. Instead of sequentially sampling actions and states, Taheri Yeganeh et al. (2024) trained multi-step latent transitions, conditioned on repeated actions; during planning, a single transition predicts the outcome while keeping an action for a fixed number of time-steps. This way, the impact of actions over a long horizon is captured using repeated-action simulations. While it can be combined with MCTS, this approximation helped distinguish different actions based on the EFE in highly stochastic control tasks with a single look-ahead (Taheri Yeganeh et al., 2024). It is limited to discrete actions, cannot go beyond repeated actions, and still requires planning via EFE computation before every action.

## 3 DEEP ACTIVE INFERENCE AGENT

From habit-integrated MCTS to hybrid-horizon and gradient-based latent imagination, state-of-the-art agents increasingly integrate policy learning with planning to capture the long-term effects essential for scalable and sample-efficient control. A prominent approach is latent imagination, notably used by Dreamer agents (Hafner et al., 2025; 2019; 2020), which perform sequential rollouts in latent space using a RSSM. Besides its computational burden, this method risks accumulating errors as networks are repeatedly inferred and sampled. These models embed the policy network in the latent space by sampling actions along each latent-state trajectory, so policy optimization depends on a large number of sampling steps in the model's imaginations.

A simpler strategy assumes a generative model that *knows* the exact form of the policy function—in other words, the network parameters themselves. We can train such a model to generate a prediction deep into the horizon with a single look-ahead, once provided with the policy parameters governing interaction with the environment over that horizon. Thus, the EFE can be computed directly over the horizon, and gradients can be backpropagated to minimize the EFE, thereby guiding the agent toward its intrinsic and extrinsic objectives. Given that the policy is optimized through the gradient steps of the EFE, this approach naturally scales to both discrete and continuous action spaces rather than choosing discrete actions, as in earlier AIF-agent implementations(Fountas et al., 2020). Here, we adopt this AIF-consistent generative-policy modeling, without incorporating further mechanisms typically employed to further enhance world-models or AIF agents.

### 3.1 ARCHITECTURE

The agent comprises, at a minimum, a **policy network** that directly interacts with the environment and a **generative model** that is trained to optimize that policy. Conditioned on the policy, the generative model constitutes the core of AIF and can be instantiated with various architectures. In this work we adopt a generic—yet commonly used—autoencoder assembly (Fountas et al., 2020) to instantiate the formalism of Sec. 2.1, which requires the tightly coupled modules illustrated in Fig. 1. Leveraging amortization (Kingma & Welling, 2013; Marino et al., 2018; Gershman & Goodman, 2014) to scale inference (Fountas et al., 2020), the generative model is parameterized by two sets: $\theta = \{\theta_s, \theta_o\}$ for prior generation and $\phi = \{\phi_s\}$ for recognition. Accordingly, the **Encoder** $Q_{\phi_s}(s_t)$ performs amortized inference by mapping the currently sampled observation $\tilde{o}_t$ to a posterior distribution over the latent state $s_t$ (Margossian & Blei, 2023). The key difference here is that, rather than sampling actions inside the latent dynamics, we incorporate a policy function—or **Actor**—$Q_{\phi_a}(a_t \mid \tilde{o}_t)$, which itself infers a distribution over actions with parameters $\phi_a$. We therefore introduce an explicit representation for the function itself with the mapping $\Pi : Q_{\phi_a} \to \hat{\pi}$, resulting in $\hat{\pi}(\phi_a)$. This approach is common in neural implicit representations (Dupont et al., 2022); recent work has moreover demonstrated that neural functions with diverse computational graphs can be embedded efficiently (Kofinas et al., 2024). Conditioned on the actor, the **Transition**, $P_{\theta_s}(s_{t+1} \mid \tilde{s}_t, \hat{\pi})$, *overshoots* the latent dynamics up to a planning horizon $H$, producing a distribution for $s_{t+H}$ given the sampled latent state at time $t$, while the actor–denoted by $\phi_a$–is assumed fixed throughout the horizon. Finally, the **Decoder** $P_{\theta_o}(o_{t+H} \mid \tilde{s}_{t+H})$ converts the predicted latent state back into a distribution over future observations. Each of the three modules in the generative model is realized by a neural network that outputs the parameters of a diagonal multivariate Gaussian, thereby approximating a pre-selected likelihood family. They can be trained end-to-end by minimizing the VFE (Eq. 1), whereas the actor is optimized—using predictions from the calibrated model—by minimizing the EFE (Eq. 4). In this way, the agent unifies the two free-energy paradigms derived in the formalism.

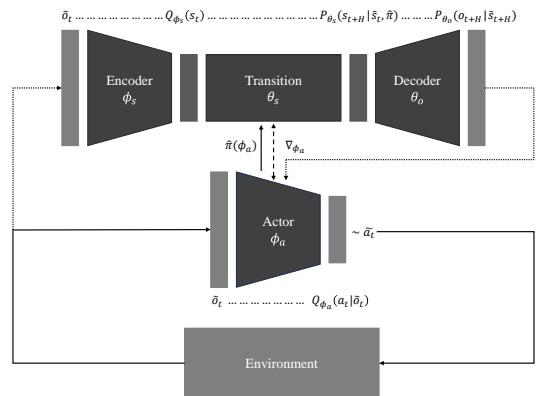

$\bar{o}_t \dots \dots \dots \dots \dots \dots Q_{\phi_s}(s_t) \dots \dots \dots \dots \dots \dots \dots P_{\theta_s}(s_{t+H}|\bar{s}_t, \hat{\pi}) \dots \dots \dots P_{\theta_o}(o_{t+H}|\bar{s}_{t+H})$

Figure 1: The Deep AIF agent architecture illustrates its interaction with the environment. The actor independently selects actions, while the generative model is used to optimize the policy.

Aside from the actor and transition, which account for latent dynamics with a single look-ahead, the architecture resembles a variational autoencoder (VAE) (Kingma & Welling, 2013); nevertheless, other generative mechanisms, such as diffusion or memory-based RSSM models, can be extended to support the same objective.

## 3.2 POLICY OPTIMIZATION

We propose a concise yet effective formulation for embedding the actor within the generative model so that it serves as a planner that minimizes the EFE via gradient descent. Conditioned on a fixed policy $\hat{\pi}(\phi_a)$, the model generates the prediction distribution $P_\theta(o_{t+H}|\phi_a)$, from which we compute the EFE, denoted as the function $G_\theta(\tilde{o}, \phi_a)$. Policy optimization then proceeds by updating the actor parameters according to the gradient $\nabla_{\phi_a} G_\theta(\tilde{o}, \phi_a)$. Most world-model agents introduce stochasticity by sampling actions during imagination, which promotes exploration—typically aided by auxiliary terms during the policy gradient. This results in a Monte Carlo estimation of the policy across imagined trajectories, which is then differentiated based on the return (Hafner et al., 2020). In contrast, our approach assumes the exact form of the policy is integrated into the dynamics, and exploration is driven by the AIF formalism based on the generative model.

To effectively estimate the different components of the EFE in Eq. 4, Fountas et al. (2020) employed multiple levels of Monte Carlo sampling. While their original formulation incorporated sampled actions over multi-step horizons, the same structure and sampling scheme remain beneficial when using an integrated actor with deep temporal overshooting. Similarly, we adopt ancestral sampling to generate the prediction $P_\theta(o_{t+H} \mid \phi_a)$ and leverage dropout (Gal & Ghahramani, 2016) in the networks. It's coupled with further sampling from the latent distributions to compute the entropies necessary for calculating the EFE terms. Crucially, under the AIF framework, agents need a prior preference over predictions to guide behavior—this is formalized through the extrinsic value (i.e., Eq. 4a). Accordingly, we define an analytical mapping that transforms the prediction distribution into a continuous preference spectrum, $\Psi : P_\theta(o_\tau) \rightarrow [0, 1]$.

Unlike RL, which relies on the return of accumulated rewards, this formulation allows the agent to express more general and nuanced forms of preference. In practice, designing a suitable reward function for RL agents remains a difficult task, often resulting in sparse or hand-crafted signals that can be costly to design and compute. The flexibility in preference, however, introduces challenges—particularly when agents have complex preference space and must act with short-sighted EFE approximations. Our approach, by optimizing planning through deep temporal prediction, mitigates this issue and enables longer-term evaluation of the extrinsic value.

### 3.2.1 TRAINING & PLANNING

During training, the generative model gradually learns how different actor parameters $\phi_a$ affect the dynamics, and during policy optimization, this learned dynamics is then used to differentiate the

actor toward lower EFE or surprise. Critical for effective policy learning is the accuracy of the world model, which forms the foundation of AIF (Friston et al., 2010; Parr et al., 2022; Fountas et al., 2020) and predictive coding (Millidge et al., 2022). To improve model training, we introduce experience replay (Mnih et al., 2015) using an experience memory/buffer $\mathcal{M}$, from which we sample batches of experiences, while ensuring that each batch includes the most recent one. We compute the VFE in Eq. 1 for these experiences to train the model with $\beta$-regularization. With the updated model, we differentiate the EFE over a batch of observations—including previous and current ones—within imagined trajectories of length $H$, training the actor similarly to world-model methods (Hafner et al., 2020; 2025; Ha & Schmidhuber, 2018). This results in a joint training algorithm 1 that alternates between updating the generative model and the policy, using the model to guide planning via policy gradients. This approach, policy learning—rather than explicit action planning—relaxes the bounded-sight constraint of EFE, as the policy is iteratively trained across diverse scenarios within the planning horizon, and its effective *sight* extends beyond the nominal horizon $H$. Recent work on AIF-based agents has also emphasized the advantages of integrating a policy network with the EFE objective (Nguyen et al., 2024). After training concludes and the agent's model is fixed, the agent can still leverage its model for planning. Specifically, EFE-based gradient updates can be applied at the observation level once every $H$ steps, effectively fine-tuning the policy for the immediate horizon.

## 4 EXPERIMENTS

Most existing AIF agents have shown effectiveness across a range of tasks typically performed by biological agents, such as humans and animals. These tasks often involve image-based observations (Nguyen et al., 2024). For example, Fountas et al. (2020) evaluated their agent on Dynamic dSprites (Higgins et al., 2016) and Animal-AI (Crosby et al., 2019), which biological agents can perform with relative ease. AIF has also been successfully applied in robotics (Lanillos et al., 2021; Da Costa et al., 2022), including object manipulation (Nguyen et al., 2024; Schneider et al., 2022), aligning with behaviors humans naturally perform. This effectiveness is largely attributed to AIF's grounding in theories of decision-making in biological brains (Parr et al., 2022). However, applying AIF to more complex domains—such as industrial system control—poses significant challenges. Even humans may struggle to design effective policies in these settings. Such environments often exhibit high stochasticity, where short observation trajectories are dominated by noise, making it difficult to optimize free energy for learning and action selection. This issue is less pronounced in world-model agents, which often use memory-based (e.g., recurrent) architectures (Hafner et al., 2020; 2025). Moreover, realistic environments frequently combine discrete and continuous observation modalities, complicating generative and sampling predictions. Delayed feedback and long-horizon requirements further challenge planning under the AIF framework. Additionally, many real-world tasks require rapid, frequent decisions and sustained performance in non-episodic, stochastic settings. We evaluate our approach in a validated, high-fidelity plant-level industrial simulator (Lofredo et al., 2023b) under the provably delayed, long-horizon setting of (Taheri Yeganeh et al., 2024). This real-world-grounded testbed provides a challenging and representative benchmark for validating the concept, demanding long-horizon planning to steer a highly stochastic class of problems toward preferred performance (see Appendix B for details).

### 4.1 RESULTS

To validate the performance of our agent in the aforementioned environment, we adopted a rigorous evaluation scheme (see Appendix D for details) based on Algorithm 1. Unlike previous works that used interactions with a batch of environments to improve training efficiency (Fountas et al., 2020), our agent was trained in each epoch by interacting with a single environment instance, reflecting a more challenging setting. The trained agent's performance was then evaluated across several randomly initialized environments. From these, the best-performing instance was selected for a one-month simulation run to assess energy efficiency and production loss, in comparison to a baseline scenario where no control was applied and machines were continuously active. We also constructed a compositional preference score—analogous to a reward function—based on time-window KPIs for energy consumption and production, serving as an overall indicator of agent performance, which is part of the observation of the agent. To enforce further regularization in the latent space to match a normal distribution, we used a *Sigmoid* function in its non-saturated domain. Since we needed

to encode the actor function, which is essentially a computational graph (Kofinas et al., 2024), we adopted a simple, non-parametric mapping $\Pi$ that concatenates the input with the first hidden and output values. Given its input–output structure and the model's continuous training on it, this mapping effectively serves as an approximation of the actor's neural function (see Appendix C for details).

We implemented the agent in the exact production system, using parameters verified to reflect realistic conditions, following the aforementioned scheme. Figure 2 presents the performance of the agent with an overshooting horizon of $H = 300$. During evaluations after each epoch (100 iterations), the agent improved the preference score of observations (Fig. 2a), which correlates with increased energy efficiency (Fig. 2b). Notably, the EFE of imagined trajectories (Fig. 2c) used for policy updates decreased as the agent learned to control the system. This trend is observed in both the extrinsic and uncertainty components of the EFE. Since policy optimization relies heavily on learning a robust generative model—with the actor integrated within it—the agent gradually improved its predictive capacity and reduced reconstruction error across both continuous (Fig. 2d, preference) and discrete (Fig. 2e,f, machine and buffer states) elements of the observation space. While EFE and overall performance eventually stabilized, the generative model continued to improve, indicating that full reconstruction of future observations is not strictly required for effective control. Finally, evaluated the trained agent over one month of simulated interaction (10 replications), applying gradient updates every $H$ steps during planning. Loffredo et al. (2023a) tested model-free RL agents (incl. DQN, PPO, and TRPO) across a reward parameter $\phi$, with DQN emerging as the top performer and close to optimal solutions. Table 1 shows DAIF agent outstrips best baseline, raising energy efficiency per production unit by $10.21\% \pm 0.14\%$ while keeping throughput loss negligible.

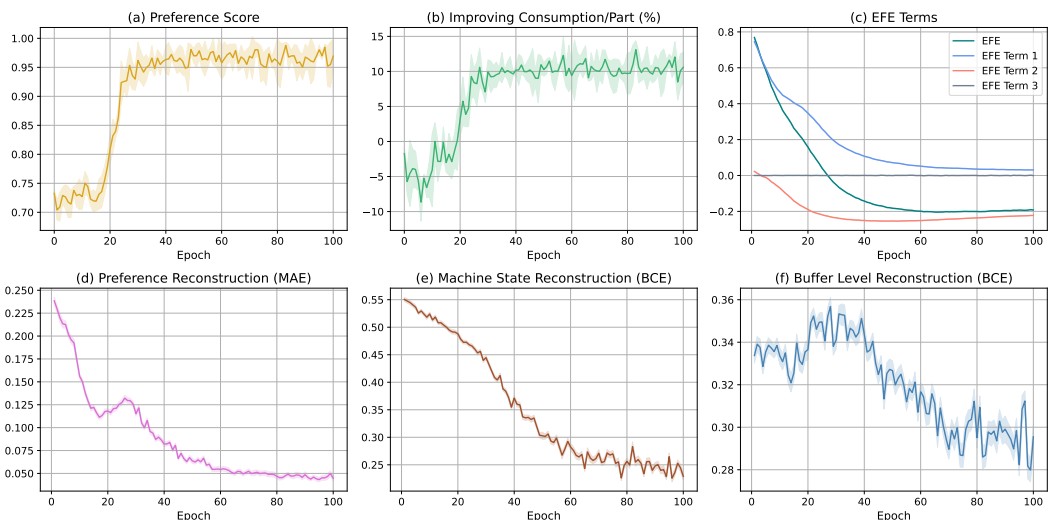

Figure 2: The performance of the agent with $H = 300$ on the real industrial system.

Table 1: Production loss versus energy savings across reward parameters $\phi$ (controlling the balance) for the best baseline agent, DQN, and the DAIF agent (with a fixed preference form).

| Agent($\phi$) | Production Loss [%] | Energy Saving [%] |
|---|---|---|
| DQN (0.93) | $4.82 \pm 0.34$ | $10.87 \pm 0.76$ |
| DQN (0.94) | $3.34 \pm 0.23$ | $9.92 \pm 0.69$ |
| **DAIF** | $\mathbf{2.59 \pm 0.16}$ | $\mathbf{12.49 \pm 0.04}$ |
| DQN (0.95) | $1.27 \pm 0.05$ | $7.00 \pm 0.07$ |
| DQN (0.96) | $1.27 \pm 0.09$ | $7.62 \pm 0.12$ |
| DQN (0.97) | $1.20 \pm 0.05$ | $7.72 \pm 0.10$ |
| DQN (0.98) | $0.54 \pm 0.04$ | $2.72 \pm 0.19$ |
| DQN (0.99) | $0.40 \pm 0.03$ | $2.46 \pm 0.01$ |

**Effect of Depth:** The agent manages to improve the performance even when the overshooting horizon can be longer (e.g., $H = 1000$ steps). We conducted experiments with different overshooting

horizons $H$ to evaluate the performance of the agent. As shown in Figure 3, we report the preference scores from the best epoch during the validation phase. We also extracted the percentage improvement in energy-efficient consumption. The results demonstrate that even with longer overshooting horizons, the agent is still able to learn robust policies.

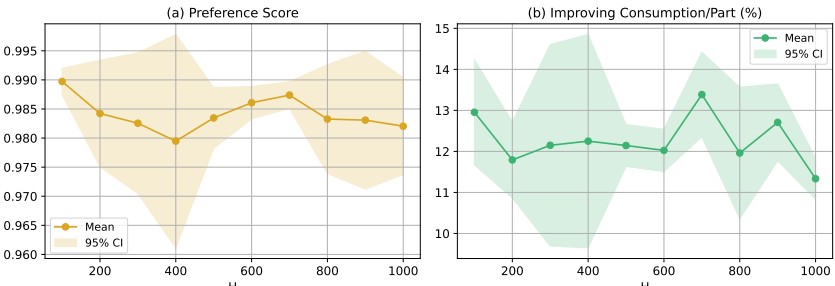

Figure 3: Performance of the agents versus overshooting horizon $H$.

## 5 CONCLUSION AND FUTURE WORK

We introduced *Deep Active Inference* (DAIF) Agents that integrate a multi-step latent transition and an explicit, differentiable policy inside a single generative model. By overshooting the dynamics to a long horizon and back-propagating expected-free-energy gradients into the policy, the agent plans without an exhaustive tree search, scales naturally to continuous actions, and preserves the epistemic–exploration balance that drives active inference. We evaluated DAIF on a high-fidelity industrial control problem whose feature complexity has rarely been tackled in previous works based on active inference. Empirically, DAIF closed the loop between model learning and control in highly stochastic, delayed, long-horizon environments. With a single gradient update every $H$ steps, the trained agent planned and achieved strong performance—surpassing model-free RL baseline—while its world model continued to refine predictive accuracy even after the policy stabilized.

**Limitations and Future Work:** While predicting an $H$-step transition removes the expensive *per-step* planning loop, the agent still has to gather *experience* after $H$ interactions and store it in the replay memory for training, so its sample efficiency can still be improved. To update the world model after each new environment interaction—reflecting the evolving actor parameters within the horizon—we need an operator that aggregates the *sequence* of actor representations. Recurrent models are a natural choice for this, but their sequential unrolling adds latency and can hinder gradient flow. A lighter alternative is to treat the $H$ embeddings as an (almost) unordered set and use a set function (Zaheer et al., 2017); when the temporal structure with simple positional embeddings (e.g. sinusoidal (Vaswani et al., 2017)) can be concatenated before the set pooling. This allows us to break the horizon into segments—down to a single step—while still enabling EFE gradient back-propagation via aggregation of the current policy representation. Finally, (neural) operator-learning techniques could enable resolution-invariant aggregation across function spaces (Li et al., 2020; Lu et al., 2021). Additional extensions include replacing the VAE world-model with diffusion- or flow-matching-based generators (Huang et al., 2024), adopting actor–critic optimization (as in Dreamer and related world-model agents (Hafner et al., 2020; 2025; Nguyen et al., 2024)), and introducing regularization schemes to stabilize EFE gradient updates and reduce their variance. Rapid adaptation in non-stationary settings—where model-free agents often struggle—remains an especially promising direction.

Overall, this work bridges neuroscience-inspired active inference and contemporary world-model RL, demonstrating that a compact, end-to-end probabilistic agent can deliver efficient control in domains where hand-crafted rewards and step-wise planning are impractical.

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

## A  DEEP ACTIVE INFERENCE

### A.1  ACTIVE INFERENCE FRAMEWORK

In summary (Fig. 4), the framework is realized through a mathematical formalism that unfolds as follows: An observation is ingested and propagated through the generative model, yielding a perceptual update—beliefs about current and future states. These beliefs enable the computation of the EFE (Eq. 4), which is used during planning to select actions. After the next observation arrives, the VFE (Eq. 1) is evaluated and used to calibrate—learn—the model by matching the new observation to the prior prediction. Each iteration optimizes the model with the VFE from the previous loop, and the updated model then guides subsequent inference for planning and action.

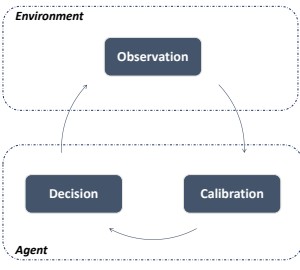 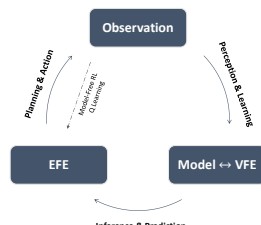

Figure 4: Two perspectives of the AIF framework: general steps (left) and core elements (right).

### A.2 ALGORITHM

## B APPLICATION

As energy efficiency becomes increasingly critical in manufacturing (Loffredo et al., 2024), RL offers a model-free alternative to traditional control, though it may struggle with rapid adaptations in non-stationary environments (Loffredo et al., 2023a). We focus on simulating workstations in an automotive manufacturing system composed of parallel, identical machines. Governed by Poisson processes for arrivals, processing, failures, and repairs (Loffredo et al., 2023b), the system evolves as a discrete-time Markov chain (Ross, 2014). Control actions—switching machines on or off—aim to improve energy efficiency without compromising throughput. The problem is continual with no terminal state, and decisions rely on both discrete and continuous observations. Due to stochastic delays, the system connects continuous-time dynamics to discrete-time decisions, making performance only observable over long horizons. Accordingly, we employ a window-based preference metric (Taheri Yeganeh et al., 2024) that evaluates KPIs over the past eight hours. The production rate is defined as $T = \frac{N(t)-N(t-t_s)}{t_s}$, where $N(t)$ is the number of parts produced, and the energy consumption rate as $E = \frac{C(t)-C(t-t_s)}{t_s}$, where $C(t)$ denotes total energy consumed, with $t - t_s \approx 8\,\mathrm{hrs}$. This window may span thousands of actions, where due to stochasticity and the integral nature of performance, immediate observations are noisy and uninformative. As a result, the AIF agents based on short-horizon EFE planning are not feasible in this setting. By operating directly on raw performance signals rather than handcrafted rewards, the approach enables scalability to domains where reward signals are sparse or expensive. The agent must handle delayed feedback and plan over extended horizons to move the system towards the preferred performance.

### B.1 ENERGY-EFFICIENCY CONTROL

Energy-Efficiency Control (EEC) is attracting growing attention in both academia and industrial research within manufacturing systems. Acting at the component, machine, and production-system levels, EEC can deliver substantial energy savings. Fundamentally, it adjusts an asset's power-consumption state to its operating context: equipment remains fully powered when its function is required and shifts to a low-power state when idle. Implementing this strategy is complicated by stochastic demand patterns and by the penalties incurred during state transitions—namely the production time lost while the asset adds no value and the energy consumed by the transition itself. A comprehensive, up-to-date survey of the field is provided in (Renna & Materi, 2021). Motivated by these, we focus on a system that replicates the key characteristics of an actual automotive manufacturing line.

### B.2 SYSTEM DESCRIPTION

Following the benchmark set by Loffredo et al. (2023b)—and readily extensible to a multi-stage production line (Loffredo et al., 2024)—we study a stand-alone workstation comprising a finite-capacity upstream buffer $B$ that feeds $c$ identical, parallel machines (Fig. 5). Parts arrive stochastically and each machine may reside in one of five states: *busy*, *idle*, *standby*, *startup*, or *failed*. The

---

**Algorithm 1** Deep AIF Agent Training (per epoch)

---

1: Initialize $\theta = \{\theta_s, \theta_o\}$, $\phi = \{\phi_s, \phi_a\}$, $\mathcal{M}$
2: Randomly initialize $E$
3: **for** $n = 1, 2, ..., N$ **do**
    ▷ ENVIRONMENT INTERACTION
4:    $\hat{\pi}_t \leftarrow \Pi(Q_{\phi_a})$
5:    **for** $\tau = t + 1, t + 2, ..., t + H$ **do**
6:      Sample a new observation $\tilde{o}_\tau$ from $E$
7:      Apply $\tilde{a}_\tau \sim Q_{\phi_a}(a_\tau | \tilde{o}_\tau)$ to $E$
8:      Sample a new observation $\tilde{o}_{\tau+1}$ from $E$
9:    $\mathcal{M} \leftarrow \mathcal{M} \cup \{(\tilde{o}_t, \hat{\pi}_t, \tilde{o}_{t+H})\}$
    ▷ MODEL LEARNING
10:   $\{(\tilde{o}_{t'}, \hat{\pi}_{t'}, \tilde{o}_{t'+H})\}^{B_1} \sim \mathcal{M}$
11:   **for** $t' = 1, 2, ..., B_1$ **do**
12:     **run** Model$(\tilde{o}_{t'}, \hat{\pi}_{t'}, \tilde{o}_{t'+H})$
13:     $\mathcal{L}_s \leftarrow \mathcal{L}_s + D_{\mathrm{KL}}\left[Q_{\phi_s}(s_{t'+H}) \,||\, \mathcal{N}(\mu, \sigma^2)\right]$
14:     $\mathcal{L}_o \leftarrow \mathcal{L}_o - \mathbb{E}_{Q(s_{t'+H})}\left[\log P_{\theta_o}(o_{t'+H} | \tilde{s}_{t'+H})\right]$
15:     $\mathcal{L}_o \leftarrow \mathcal{L}_o + \beta * D_{\mathrm{KL}}\left[Q_{\phi_s}(s_{t'+H}) \,||\, \mathcal{N}(\tilde{\mu}, \tilde{\sigma}^2)\right]$
16:   $\theta_s \leftarrow \theta_s - \xi \nabla_{\theta_s} \mathbb{E}\left[\mathcal{L}_s(\theta_s)\right]$
17:   $\phi_s \leftarrow \phi_s - \gamma \nabla_{\phi_s} \mathbb{E}\left[\mathcal{L}_s(\phi_o)\right]$
18:   $\theta_o \leftarrow \theta_o - \eta \nabla_{\theta_o} \mathbb{E}\left[\mathcal{L}_o(\theta_o)\right]$
    ▷ POLICY OPTIMIZATION
19:   $\{\tilde{o}_\tau\}^{B_2} \sim \mathcal{M}$
20:   **for** $\tau = 1, 2, ..., B_2$ **do**
21:     Compute $Q_{\phi_s}(s_\tau)$ using $\tilde{o}_\tau$
22:     Sample $\tilde{s}_\tau \sim Q_{\phi_s}(s_\tau)$
23:     **for** $s = 1, 2, ..., S_1$ **do**
24:       Compute $\mu, \sigma \leftarrow P_{\theta_s}(s_{\tau+H} | \tilde{s}_\tau, \hat{\pi}_t)$
25:       Sample $\tilde{s}_{\tau+H} \sim \mathcal{N}(\mu, \sigma^2)$
26:       Compute $P_{\theta_o}(o_{\tau+H} | \tilde{s}_{\tau+H})$
27:       Compute $Q_{\phi_s}(\tilde{s}_{\tau+H})$ using $\tilde{o}_{\tau+H}$
28:       Compute $\mu', \sigma' \leftarrow Q_{\phi_s}(\tilde{s}_{\tau+H})$
29:       $G \leftarrow G - \log \Psi\left[P_{\theta_o}(o_{\tau+H} | \tilde{s}_{\tau+H})\right]$
30:       $G \leftarrow G + \left[H(\mu', \sigma') - H(\mu, \sigma)\right]$
31:       **for** $s = 1, 2, ..., S_2$ **do**
32:         Sample $\tilde{s}_{\tau+H} \sim P_{\theta_s}(s_{\tau+H} | \tilde{s}_\tau, \hat{\pi}_\tau)$ ▷ *Re-computed with dropout.*
33:         Compute $\mu'', \sigma'' \leftarrow P_{\theta_o}(o_{\tau+H} | \tilde{s}_{\tau+H})$
34:         Sample $\tilde{s}_{\tau+H} \sim \mathcal{N}(\mu, \sigma^2)$
35:         Compute $\mu''', \sigma''' \leftarrow P_{\theta_o}(o_{\tau+H} | \tilde{s}_{\tau+H})$
36:         $G \leftarrow G + \left[H(\mu'', \sigma'') - H(\mu''', \sigma''')\right]$
37:   $\phi_a \leftarrow \phi_a - \alpha \nabla_{\phi_a} \mathbb{E}\left[G(\phi_a)\right]$

---

**Agent components:**
  Model:
    Encoder $Q_{\phi_s}$.
    Transition $P_{\theta_s}$.
    Decoder $P_{\theta_o}$.
  Actor $Q_{\phi_a}$.
  Actor mapping $\Pi$.
  Preference mapping $\Psi$.

**Other components:**
  Environment $E$.
  Experience Memory $\mathcal{M}$.

**Hyperparameters:**
  Iterations $N$.
  Beta $\beta$.
  Horizon $H$.
  Batch size $B_1$, $B_2$.
  Sample size $S_1$, $S_2$.
  Learning rate $\xi$, $\gamma$, $\eta$, $\alpha$.

**Run** Model$(\tilde{o}_i, \hat{\pi}, \tilde{o}_{i+H})$:
  Compute $Q_{\phi_s}(s_i)$ using $\tilde{o}_i$
  Sample $\tilde{s}_i \sim Q_{\phi_s}(s_i)$
  Compute $\mu, \sigma \leftarrow P_{\theta_s}(s_{i+H} | \tilde{s}_i, \hat{\pi})$
  Compute $Q_{\phi_s}(\tilde{s}_{i+H})$ using $\tilde{o}_{i+H}$
  Compute $\mu', \sigma' \leftarrow Q_{\phi_s}(\tilde{s}_{i+H})$
  Sample $\tilde{s}_{i+H} \sim \mathcal{N}(\mu, \sigma^2)$
  Compute $P_{\theta_o}(o_{i+H} | \tilde{s}_{i+H})$

---

corresponding power rates satisfy:

$$w_b > w_{su} > w_{id} > w_{sb} \approx w_f \approx 0.$$

All system processes are modeled as Poisson processes (Kingman, 1992); this pertains to the arrival rate ($\lambda$) to buffer $B$ with capacity $K$, machine processing times ($\mu$), startup times ($\delta$), time between failures ($\psi$), and time to repair ($\xi$), all with expected values, independent and stationary. Table 2 summarizes the parameters used to replicate the real industrial case study reported in (Loffredo et al., 2023b).

Each machine processes a single part type under a first-come-first-served policy. Machines cannot be powered down while processing or during startup. If a machine is ready to work but the buffer $B$ is empty, it becomes *starved* and enters the idle state. The central challenge of EEC in this system is to dynamically determine how many machines should remain active versus how many should be

Figure 5: Layout of parallel, identical machines in the workstation (Loffredo et al., 2023b).

Table 2: Parameters for replicating the industrial system (Loffredo et al., 2023b).

| Parameter | $c$ | $K$ | $\mu$ | $\delta$ | $\psi$ | $\xi$ |
|---|---|---|---|---|---|---|
| Value | 6 | 10 | 0.012 | 0.033 | 0.001 | 0.033 |
| Parameter | $\lambda$ | $w_b$ | $w_{id}$ | $w_{su}$ | $w_{sb}$ | $w_f$ |
| Value | 0.050 | 15 kW | 9.30 kW | 10 kW | 0 kW | 0 kW |

transitioned to low-power states. This decision must be made adaptively in response to the unfolding stochastic conditions, striking an optimal balance between reducing energy usage and maintaining a high production rate (i.e., throughput).

### B.2.1 MODELING

As machine state transitions and part arrivals are modeled as Poisson processes (Kingman, 1992; Loffredo et al., 2023b), we adopt the *event-driven* scheme of Loffredo et al. (2023b;a), where control decisions are triggered immediately after each state change in the system rather than at fixed sampling intervals—an approach proven effective for managing active machines. In this way, the system itself requests decisions from the agent in a stochastic manner.

This control task admits two different formulations (Taheri Yeganeh et al., 2024): (i) continuous-time stochastic control or (ii) a discrete-time Markov chain (DTMC) (Ross, 2014). A continuous-time model must provide the raw inter-event interval $\Delta t$ to the agent for every machine and subsequent observation, whereas the DTMC abstraction lets the agent infer transition probabilities directly from observed events. Because $\Delta t$ varies from event to event, a continuous-time formulation would have to align the predictor $P_\theta(o_{t+H})$ with the reference observation $\tilde{o}_{t+H}$, which—although beneficial for planning—complicates the network architecture owing to state occupancy durations. Therefore, to keep the model simpler, we adopt the discrete-time, *event-driven* formulation.

### B.3 PREFERENCE MAPPING

In AIF, the agent acts to reach its preferred observation, akin to a setpoint in control theory (Friston et al., 2017; Millidge et al., 2022). This implies that the agent possesses an internal preference mapping $\Psi$, which quantifies how close its predicted observation is to a desired target. While conceptually related to reward functions in RL, this preference reflects a control-based objective rather than cumulative rewards in the Markov Decision Process framework (Sutton & Barto, 2018).

Building on the EEC framework introduced by Loffredo et al. (2023a), a generic preference or reward function for the multi-objective optimization of the system under study can include terms for production, energy consumption, and a weighted combination thereof (Taheri Yeganeh et al., 2024):

$$R_{\text{production}} = \frac{T_{\text{current}}}{T_{\text{max}}}, \quad R_{\text{energy}} = 1 - \frac{E_{\text{avg}}}{E_{\text{max}}}, \tag{7a}$$

$$R = \phi \cdot R_{\text{production}} + (1 - \phi) \cdot R_{\text{energy}}, \tag{7b}$$

where $\phi \in [0, 1]$ is a weighting coefficient balancing the importance of production and energy efficiency.

Loffredo et al. (2023a) computed the production term as the average throughput from the start of the interaction, and the energy term as the difference between consecutive time steps, followed by exponential transformations. In contrast, we employ a window-based approach (Taheri Yeganeh et al., 2024), which better captures the stochastic performance of the system and aligns with the concept of a *delayed* and *long-horizon* control problem. Specifically, we evaluate average system performance over a fixed time span $t_s$[1], leading up to the current observation at time $t$. Accordingly:

$$T_{\text{current}} = \frac{NP(t) - NP(t - t_s)}{t_s},$$

$$E_{\text{avg}} = \frac{C(t) - C(t - t_s)}{t_s},$$

where $NP(t)$ is the number of parts produced and $C(t)$ is the total energy consumed up to time $t$. $T_{\max}$ corresponds to the maximum achievable throughput under the *ALL ON* policy (Loffredo et al., 2023a), and $E_{\max}$ denotes the theoretical peak energy consumption when all machines operate in the busy state.

To encourage EEC, $\phi$ is typically set close to 1 to avoid excessive production loss. However, this linear formulation may overestimate performance in cases where energy savings are negligible—i.e., the composite term remains high due to production alone, even when control is not applied. To address this, we adjust the preference function by applying a sigmoid transformation to the energy term:

$$R = R_{\text{production}} \cdot \sigma(c_r R_{\text{energy}}), \tag{8}$$

where $\sigma(x) = \frac{1}{1+e^{-x}}$ is the sigmoid function, and $c_r$ is a scalar hyperparameter controlling the sensitivity to energy savings. This formulation ensures that energy savings sharply amplify the preference, thereby enforcing a balanced focus on both productivity and energy saving. Notably, in the absence of any control actions (i.e., under the *ALL ON* policy), the energy term saturates near zero, and the composite term is naturally lower than that of the linear formulation in Equation 7.

## C  AGENT

### C.1  SETUP

For the representation of the actor function, we adopted a simple approximating mapping $\Pi$, which concatenates the input with both the first hidden layer and the output values. In this way, the policy parameters are introduced within the generative model and optimized via gradient descent on the EFE, while the rest of the agent parameters are kept fixed. We adhere to the Monte Carlo sampling methodology for calculating the EFE as outlined by Fountas et al. (2020). We also achieved similar control performance by computing all EFE terms using single-loop forward passes of the generative model repeated multiple times, which were faster and less computationally demanding than the multi-loop scheme presented in the algorithm.

Bernoulli and Gaussian distributions are employed to model the prediction and state distributions, respectively. We also regularize the state space by applying non-linear activation functions to both the encoder and transition networks. The outputs of these networks define the means and variances of Gaussian-distributed latent states. Specifically, we use the *tangent hyperbolic* (`tanh`) function for the means, and the *sigmoid* function, scaled by a factor $\lambda_s \in [1, 2]$[2], for the variances. This combination enforces additional regularization and contributes to the stability of the latent state space, ensuring values remain bounded and well-suited to a normal distribution.

### C.1.1  OBSERVATION

Given that the problem under study includes *discrete* and *continuous* elements (as we also include *preference scores* in the system states), it has a composite format; at each decision step $t$ the agent observes $o^{(t)} = \left[ o_b^{(t)}, o_m^{(t)}, o_r^{(t)} \right]$, where $o_b^{(t)}$: *discrete* is one–hot buffer-occupancy indicator, $o_m^{(t)}$:

---

[1] Eight hours or one shift in our implementation.

[2] We set $\lambda_s = 1.5$ to ensure the variance output remains within the non-saturated domain of the sigmoid function, thereby preserving informative gradient flow during training.

*discrete* is one–hot machine-state indicators, and $o_r^{(t)}$: *continuous* real-valued preference scores, includes production, energy, composite terms. Similarly, the generative model outputs the corresponding prediction $P(o^{(t)}) = [P_b, \ P_m, \ P_r]$, with all components generated with Bernoulli parameters. As the preference elements contain a continuous component, during EFE computation we need the entropy of the predicted observation distribution. To keep the procedure analytically tractable and consistent with the binary part of the observation, we approximate these continuous preference outputs with Bernoulli-like parameters; i.e. we treat each scalar reward prediction $P_r \in [0, 1]$ as if it were the mean of a Bernoulli variable when evaluating the entropy term. In practice, this is an approximation $P_r$ already lies in $[0, 1]$—yet allows us to reuse the same closed-form binary-entropy expression for both discrete and continuous preference channels to ease the computation.

When we calculate the third term of the EFE, we need to feed the prediction to the encoder. We sample[3] the one-hot parts first and then apply them to the encoder. This differs from earlier AIF implementations such as Fountas et al. (2020), which feed *mean pixel intensities* (i.e. predicted Bernoulli parameters in $[0, 1]$) straight back into the encoder. That works for images, but in our setting (one-hot vectors) it would (i) ignore the semantics of one-hot codes and (ii) treat each feature independently, discarding correlations.

### C.1.2 RECONSTRUCTION LOSS

Based on the format of the observation and the respective prediction, we need to distinguish the reconstruction loss. Accordingly, for a mini-batch of size $N$ we compute

$$\text{BCE}_b = \frac{1}{N} \sum_{i=1}^{N} \text{BCE}\left(P_b^{(i)}, \ o_b^{(i)}\right) \tag{9a}$$

$$\text{BCE}_m = \frac{1}{N} \sum_{i=1}^{N} \text{BCE}\left(P_m^{(i)}, \ o_m^{(i)}\right) \tag{9b}$$

$$\text{MAE}_r^{(i)} = \left| P_r^{(i)} - o_r^{(i)} \right| \tag{9c}$$

$$\text{MSE}_r = \frac{1}{N} \sum_{i=1}^{N} \left[ -\log\left(1 - \text{MAE}_r^{(i)} + \varepsilon\right) \right] \tag{9d}$$

This combines binary cross-entropy (BCE) with an exponential-like term for the continuous elements. Given that preference is generally more important than the other elements, and that buffer level is more important than the machine state, we weight these elements. Finally, it is combined with the usual $\beta$-VAE regularization term and passed to the optimizer as:

$$\mathcal{L}_o = \underbrace{\frac{2}{7}\text{BCE}_b + \frac{1}{7}\text{BCE}_m + \frac{4}{7}\text{MSE}_r}_{\text{reconstruction}} + \underbrace{\beta * D_{\text{KL}}\big(q(s) \, \| \, \mathcal{N}(\mathbf{0}, \mathbf{I})\big)}_{\beta \text{ regularization}}. \tag{10}$$

## D EXPERIMENTS

### D.1 CODE

The code and data are available at the anonymous repository.

### D.2 TRAINING AND EVALUATION PROCEDURE

During each training epoch, an environment is initialized with the parameters of the industrial system, including stochastic processes. The system first undergoes a one-day warm-up period in simulation time using the *ALL ON* policy. The resulting profile from this warm-up is discarded. This is

---

[3]For ease of calculation, we take the $\max$ for the states.

followed by another one-day simulation using a *random* policy to bring the system to a fully random and uncontrolled state. Then, the agent, equipped with experience replay, interacts with the system. During each epoch, the model is trained for several iterations, with updates both for the model and actor occurring every $H$ steps, following sampling from the experience memory. After each epoch, the agent's performance is validated on three independent and randomly instantiated environments that undergo the same warm-up and random initialization steps. These validation episodes span one day of simulation, during which no model learning or experience gathering occurs, except for gradient fine-tuning of the actor every $H$ steps. The performance is then averaged, and the standard deviation is computed. While all previous performance metrics are based on the preference score, the best-performing agent during validation is retrieved and tested for 30 days of simulation on 10 independent and randomly instantiated environments. These environments follow the same initialization protocol, except for a 10-day warm-up to ensure consistency. Final performance of relative energy efficiency is averaged over the 10 environments, along with the computation of standard deviations.

