# OpenReview forum: "Deep Active Inference Agents for Delayed and Long-Horizon Environments"
_ICLR.cc/2026/Conference — ICLR 2026 Conference Withdrawn Submission_

### Official Review · Reviewer_yExp · 2025-10-28

**Soundness:** 2
**Presentation:** 1
**Contribution:** 2
**Rating:** 2
**Confidence:** 3

**Summary:**

This paper focuses on addressing the long-horizon world model and its application to realistic scenarios.

They introduce Deep Active Inference (DAIF), a world-model agent that predicts latent dynamics H steps ahead and embeds the policy within the generative model, allowing expected free energy (EFE) gradients to update the actor directly.

The model combines an encoder–transition–decoder with an explicit actor $\Pi(Q_{\phi_a}) \to \hat{\pi}(\phi_a)$; the transition $ P_ {\theta_ s}(s_ {t+H} \mid \tilde{s}_ t,\hat \pi) $ predicts H-step futures, and the decoder reconstructs $o_{t+H}$. Training uses VFE/ELBO for the model and EFE-based gradient updates for the actor, with Monte Carlo sampling, dropout, and a replay buffer. After training, policy planning applies a single EFE gradient step every H steps.

Experiments on an event-driven industrial energy-efficiency control task (discrete states/actions) show that DAIF outperforms a DQN baseline, achieving greater energy savings with minimal throughput loss across H ∈ {200–1000}.

**Strengths:**

Conditioning the H-step transition on an explicit actor representation allows single‑shot look‑ahead and direct EFE gradient updates to the policy, cleanly tying together VFE (model learning) and EFE (policy learning). The joint training on the actor is impressive.

The work targets delayed, long‑horizon settings that are indeed challenging for short‑sighted EFE agents, and demonstrates feasibility at H up to 1000 (Figure 3).

An event-driven production system with Poisson failures/repairs and energy-throughput trade-offs is a valuable testbed; the preference is defined directly on KPI windows without hand-crafted rewards. Figure 5 and Table 2 in the appendix help ground the setting.

On this benchmark, DAIF improves energy saving with minimal throughput loss vs. DQN (Table 1, p. 8), while EFE terms and reconstruction errors trend in the expected direction during training (Figure 2, p. 8).

**Weaknesses:**

The paper lacks comparisons to world‑model agents, AIF+MCTS or the repeated‑action multi‑step AIF planner it cites as related. This makes it hard to attribute gains to the proposed architecture versus known alternatives for long‑horizon planning.

The paper claims natural scaling to continuous actions, but the single task uses discrete on/off control; no continuous‑action experiments are reported.

While the paper repeatedly emphasizes that DAIF “handles delayed and long-horizon environments” and predicts dynamics “up to H = 1000 steps,” the experimental validation is limited to a single industrial control simulator with discrete on/off actions and no direct long-horizon baseline comparisons (e.g., Dreamer, EfficientZero, or repeated-action AIF).

Presentation: The presentation of this paper is unclear to me. I cannot grasp the main points from the captions of the figures and tables. The methodology section is not well-presented, making it difficult to understand the key contributions. Additionally, the introduction does not effectively guide the discussion or highlight the significance of the research problem.

**I appreciate the idea behind this paper; however, it appears to have been done in a short time with unpolished details and writing. I would consider raising the score if the author's rebuttal clarifies these points.**

**Questions:**

Please clarify the discrepancy between §4.1 and Appendix D.2. Was the best agent (not environment) selected from validation and then evaluated on 10 independent seeds for 30 days?

Can you add Dreamer‑style (value‑gradient) and AIF+MCTS baselines, and the repeated‑action multi‑step AIF planner (Taheri Yeganeh et al., 2024), on the same environment and preference mapping? This would situate DAIF’s gains more clearly

Why concatenate input + first hidden + output rather than, say, last hidden or a learned projection? Can you provide an ablation comparing several $\Pi$ choices and an analysis of how sensitive the transition model is to small policy changes?

Could figure 2 include more details and make the text a little bit larger?

---

### Official Review · Reviewer_4UqV · 2025-10-30

**Soundness:** 2
**Presentation:** 1
**Contribution:** 1
**Rating:** 0
**Confidence:** 3

**Summary:**

This paper proposes Deep Active Inference (DAIF), an active inference framework for environments with delayed and long-horizon feedback. The main idea is to (1) learn a latent recurrent dynamics model to predict long-horizon rollouts (2) jointly update the policy in-the-loop using the gradients of the expected free energy (EFE) estimated by the model with respect to the policy. The expected free energy is estimated as the negative surprise of the future observation relative to the preferred observation (effective reward), plus the latent state epistemic uncertainty and the parameter epistemic uncertainty. This framework enables the policy to update continuously while interacting with a single environment instance. The paper evaluates the proposed method in a simulated energy production task tuned to a real-world production system. The method is shown to improve the energy efficiency of production units.

**Strengths:**

1. The paper takes a rather unique perspective on model-based reinforcement learning. Instead of planning with a learned dynamics model or employing a separate policy extraction loop, Deep Active Inference amortizes policy optimization into deployment time, performing one-step gradient descent in the loop. This is closely relevant to the continual learning literature.
2. The proposed method is grounded in the rich mathematical framework of active inference.
2. The evaluation on energy production units has a real-world impact.

**Weaknesses:**

1. The paper is poorly written. The sections are dense and lack clarity. There isn't a clear separation between related work and the actual proposed method, as everything is interleaved. The paper also fails to highlight the key contribution that makes it work in delayed and long-horizon environments. The paper needs extensive revision to be considered for publication.
2. While evaluation on energy production has a real-world impact, it is of little interest to the community.
3. The proposed method would not scale to high-dimensional observations (e.g., images). Specifically, equation 4(c) requires estimating the epistemic uncertainty of the parameters, which is intractable for large neural networks.

**Questions:**

1. Can you explain how the preference is incorporated into the expected free energy term, so that by minimizing the free energy, we achieve the preferred observations? From my understanding, the term $\log P(o_\tau | \pi)$ only maximizes the likelihood of the future observation under the model. Does the model assign a higher likelihood to the preferred observations?
2. The abstract states that "robotics and vision benchmarks ... fall short of real-world industrial complexity." This statement is not true. I wouldn't say the energy production problem from low-dimensional observations is more complex than robotics manipulation from image observations.
3. In Equation (1), Q(s, a) is not defined. This makes the formalism section difficult to parse.
4. Typo: line 394, missing "Finally, [we] evaluated..."

---

### Official Review · Reviewer_g1uo · 2025-10-31

**Soundness:** 2
**Presentation:** 1
**Contribution:** 2
**Rating:** 2
**Confidence:** 3

**Summary:**

The paper proposes a Deep Active Inference agent that integrates a policy representation into the world model so that the transition model can “overshoot” directly to an H-step prediction under a fixed policy representation. The agent then computes EFE over the long-horizon prediction and backpropagates its gradients into the actor, replacing per‑step planning with a single gradient update every H steps.

**Strengths:**

- The paper carefully lays out VFE/EFE and their decomposition, then ties EFE to the planning objective.
- The main idea, conditioning the transition on an explicit representation of the actor so a single look‑ahead reaches $t{+}H$, is interesting and potentially useful when exhaustive planning is too slow.
- Evaluating on a realistic, delayed, stochastic control task is a meaningful contribution.

**Weaknesses:**

### 1) Novelty
The specific novelty here is the policy‑conditioned H-step transition that predicts under a fixed policy representation, rather than sampling actions step‑by‑step in latent space. That distinction should be emphasized, scoped, and compared against prior “policy‑aware” or “goal/policy‑conditioned” dynamics. As written, the claim “eliminating exhaustive planning” risks over‑claiming relative to gradient‑based latent imagination used by prior agents.

### 2) Modeling
The method relies on a VAE‑like model with Gaussian latents for mixed type observations; continuous KPIs are handled by a Bernoulli‑style entropy approximation during EFE computation. The authors wrote "in practice, designing a suitable reward function for RL agents remains a difficult task, often resulting in sparse or hand-crafted signals that can be costly to design and compute," but are the VAE-like architecture and Bernouilli assumption a restricted limitation too?

### 3) Experiments
- The paper selects the best‑performing instance from validation for the one‑month run. That can inflate reported gains; standard practice is to evaluate some seeds or a fixed selection protocol (e.g., the last k checkpoints.)

- If the main motivation is single look‑ahead instead of exhaustive planning, we need wall‑clock and environment‑step comparisons to a strong MBRL baselines.

- The text claims the approach “naturally scales to continuous actions,” but experiments are discrete on/off control. A simple continuous‑action benchmark (even simulated) would substantiate the claim.

### 4) Writing
Much of section 2 could be split into concise Related Work (Dreamer/PlaNet/MuZero/EfficientZero, AIF agents, hybrid‑horizon) vs Background. This would reduce repetition (e.g., repeated references to Fountas 2020). Dropout, and sigmoid energy shaping are implementation choices which could be moved to the appendix.

**Questions:**

- Please formalize the joint distribution and make it explicit that $P_{\theta_s}(s_{t+H} \mid \tilde{s}_t, \hat{\pi}(\phi_a))$ treats $\phi_a$ as fixed over the horizon. How does training handle the fact that $\phi_a$ is changing over epochs?

- Since the composite preference $R$ enters the observation vector and the decoder predicts it, is there any double counting when $\Psi$ maps predictions to extrinsic value?

- Why not include comparisons to Dreamer‑style and a recent AIF agents? Also, in Table 1 are the DQN values are taken from prior work? If yes, were hyperparameters re‑tuned in your setting? Are you also running them for a month?

- What is the wall‑clock overhead of computing EFE gradients every H steps?

---

### Official Review · Reviewer_riQS · 2025-11-01

**Soundness:** 3
**Presentation:** 3
**Contribution:** 1
**Rating:** 2
**Confidence:** 4

**Summary:**

This paper a active inference (AI) framework to enable efficient long-horizon decision making. The authors propose a generative model that predicts an entire horizon in a single step and an integrated policy network that enables the transition
and receives gradients of the expected free energy. The method is evaluated in a realistic industrial environment with delayed feedback, and the approach achieves effective, end-to-end probabilistic control over long horizons.

**Strengths:**

The paper is fairly well written and easy to follow. The problem tackled in the paper of delayed/sparse environments is very important/relevant, and the paper proposes an orthogonal paradigm to the most common approaches for RL/MBRL.

**Weaknesses:**

There are papers such as https://arxiv.org/pdf/2009.01791 that also study perception and action as divergence minimization. This paper is not mentioned at all. Moreover, as far as I can tell, the main novelty of the paper is the generative dynamics model and how it is integrated into policy planning. However, the method is not compared against Dreamer itself, which I would say is the most obvious baseline. Generally, the baselines considered in the paper are quite outdated, combined with the fact that the environment used in the experiments is also non-standard. I am not quite sure how the paper compares against the SOTA, or rather, what is the empirical benefit of what the authors propose over the more traditional methods. Finally, several papers such as (https://arxiv.org/pdf/2005.05960, https://openreview.net/pdf?id=VGdqa79ugx) also propose maximizing epistemic uncertainty in the latent space of a world model. The latter one actually proves that this corresponds to optimistic exploration and yields sublinear regret. These methods are not discussed at all in the paper.

**Questions:**

Why is Dreamerv3 or v4 not evaluated in the paper? How would the proposed method compared against Dreamer and what would be the most crucial benefit of the method over Dreamer?

---

### Note · Authors · 2025-12-04

**Comment:**

We thank the reviewers for their time and all the program committee members for their efforts. These collective contributions help advance the field.

**Withdrawal Confirmation:**

I have read and agree with the venue's withdrawal policy on behalf of myself and my co-authors.